# Decasubstituted Pillar[5]arene Derivatives Containing *L*-Tryptophan and *L*-Phenylalanine Residues: Non-Covalent Binding and Release of Fluorescein from Nanoparticles

**DOI:** 10.3390/ijms24097700

**Published:** 2023-04-22

**Authors:** Vildan Sultanaev, Luidmila Yakimova, Anastasia Nazarova, Olga Mostovaya, Igor Sedov, Damir Davletshin, Elvina Gilyazova, Emil Bulatov, Zhang-Ting Li, Dan-Wei Zhang, Ivan Stoikov

**Affiliations:** 1A.M. Butlerov Chemistry Institute, Kazan Federal University, 18 Kremlyovskaya Str., 420008 Kazan, Russia; vildan_sultanaev@mail.ru (V.S.);; 2Institute of Fundamental Medicine and Biology, Kazan Federal University, 18 Kremlyovskaya Str., 420008 Kazan, Russia; 3Department of Chemistry, Shanghai Key Laboratory of Molecular Catalysis and Innovative Materials, Fudan University, 2205 Songhu Road, Shanghai 200438, China; 4Federal State Budgetary Scientific Institution «Federal Center for Toxicological, Radiation, and Biological Safety», Nauchny Gorodok, 2, 420075 Kazan, Russia

**Keywords:** pillar[5]arene, *L*-Tryptophan, *L*-Phenylalanine, fluorescein, nanoparticles, self-assembly, dye release

## Abstract

Sensitive systems with controlled release of drugs or diagnostic markers are attractive for solving the problems of biomedicine and antitumor therapy. In this study, new decasubstituted pillar[5]arene derivatives containing *L*-Tryptophan and *L*-Phenylalanine residues have been synthesized as pH-responsive drug nanocarriers. Fluorescein dye (**Fluo**) was loaded into the pillar[5]arene associates and used as a spectroscopic probe to evaluate the release in buffered solutions with pH 4.5, 7.4, and 9.2. The nature of the substituents in the pillar[5]arene structure has a huge influence on the rate of delivering. When the dye was loaded into the associates based on pillar[5]arene derivatives containing *L*-Tryptophan, the **Fluo** release occurs in the neutral (pH = 7.4) and alkaline (pH = 9.2) buffered solutions. When the dye was loaded into the associates based on pillar[5]arene with *L*-Phenylalanine fragments, the absence of release was observed in every pH evaluated. This happens as the result of different packing of the dye in the structure of the associate. This fact was confirmed by different fluorescence mechanisms (aggregation-caused quenching and aggregation-induced emission) and association constants. It was shown that the macrocycle with *L*-Phenylalanine fragments binds the dye more efficiently (lgK_a_ = 3.92). The experimental results indicate that the pillar[5]arene derivatives with amino acids fragments have a high potential to be used as a pH-responsive drug delivery devices, especially for promoting the intracellular delivering, due to its nanometric size.

## 1. Introduction

Pillar[5]arenes are a promising class of molecular receptors [1,2] that were first synthesized in 2008 [3]. Their distinctive feature is the presence of planar chirality (the chirality plane) (Figure 1). The unique properties of the macrocycle such as chirality [4], cavity size [5], highly symmetrical structure [6], and others attract the attention of researchers from different fields [7].

Chiral amino acids are often used to functionalize macrocyclic platforms to create biomimetic receptors. Up to date, the literature presents a number of works dedicated to the introduction of amino acid fragments, both aromatic and aliphatic, into the structure of pillar[5]arenes [8,9]. The introduction of aromatic amino acid substituents into the structure of pillar[5]arenes make it possible to fix a certain conformation, increase solubility in water, and the number of additional interaction centers to improve the receptor properties of macrocycles. Amino acids are essential building blocks in living organisms. They are involved in the miscellaneous processes of biosynthesis and metabolism and have an affinity for a large number of organic substrates. The combination of pillar[5]arenes supramolecular platform with amino acid fragments leads to the widespread application of compounds based on them in various fields.

Among a number of aromatic amino acids, tryptophan and phenylalanine are vital and have a significant impact on macrocycle geometry and self-assembly properties [10]. Fragments of phenylalanine often tend to form layered structures due to the hydrogen bonds stabilized by Van der Waals forces [11]. Tryptophan has a stabilizing effect on bioactive molecules as well as active fluorescent properties, and contributes to molecular recognition and the binding of guests through aromatic, hydrogen, and π–π interactions [12,13].

Fluorescein and its derivatives, known as xanthene dyes, are well studied and widely used in a diversity of fields. For example, fluorescein-linked glycosides showed high antioxidant activity [14] and sensitivity to changes in the acidity of the medium [15]. Fluorescein derivatives are also widely used as bio and chemosensors [16].

Fluorescein-based nanoparticles, in which the dye acts as a drug-like model, are especially attractive for study [17,18]. Such nanometer particles can be used as transport systems for the delivery of therapeutic agents and synthetic gene vectors [19]. Fluorescein in the most fluorescein-based nanoparticles is covalently linked to a substrate or metal ion, which undoubtedly limits the use of these systems in certain areas [20,21]. Therefore, an important task is the creation of nanoparticles through non-covalent self-assembly, where one of the initial building blocks is a macrocyclic derivative with an advanced structure.

In this work, it was proposed to develop an approach for obtaining nanoparticles based on fluorescein and pillar[5]arene derivatives containing *L*-Tryptophan (*L*-Trp) and *L*-Phenylalanine (*L*-Phe) residues on both rims, with the identification of the influence of the substituent nature in the macrocycle structure on the properties of the resulting particles. In this regard, the aims of this work were the synthesis of decasubstituted pillar[5]arene derivatives with amino acid residues (*L*-Trp and *L*-Phe) and their use to obtain fluorescein containing nanoparticles with different fluorescence, and further study of the possibility of pH-dependent dye release from nanoparticles.

## 2. Results and Discussion

### 2.1. Synthesis of Compounds ***2***, ***3***

To create fluorescein-containing nanoparticles with a different stability and mode of interaction, two macrocycles were chosen: decasubstituted pillar[5]arene derivatives containing positively charged quaternary nitrogen atoms, and amide and amino acid moieties of *L*-Tryptophan (**2**) and *L*-Phenylalanine (**3**) on both rims in the structure.

In connection with this, the first step of the work was the synthesis of highly reactive alkylating agents based on *L*-Tryptophan and *L*-Phenylalanine [22] for their further introduction into the macrocycle structure according to Menshutkin reaction.

Macrocycles **2** and **3** containing ester fragments of *L*-Trp and *L*-Phe, respectively, were synthesized. Reactions were carried out between amine **1** obtained in 81% yield according to the literature method [23] and the corresponding alkylating agent. The mixtures were refluxed in acetonitrile within 12–15 h. Then, the solvent was removed under reduced pressure, and target products were recrystallized from propanol-2 (Figure 1). pillar[5]arenes **2**, **3** were obtained in high 82% and 80% yields, respectively.

Pillararenes **2** and **3** have asymmetric carbon atoms in the phenylalanine and tryptophan fragments. The reaction can proceed stereospecifically with the predominance of one of the conformers when the initial decamine **1** interacts with alkylating agents with chiral centers. However, earlier, our research group established that in this case the reaction products exist not as racemic mixtures but as a mixture of two conformers with the predominance of one of them [23].

Perfunctionalization is a suitable way to control the solubility properties of the initial water insoluble pillar[5]arene [24]. The construction of macrocycles with the introduction of positively charged quaternary nitrogen atoms, amide, and moderately water-soluble amino acid fragments should have a positive effect on its solubility in water [25]. It was expected that macrocycles with amino acid fragments of different hydrophobicity would have a different solubility in water. Thus, out of the two synthesized pillar[5]arenes, only the macrocycle **3** turned out to be water soluble, while pillar[5]arene **2** with *L*-Tryptophan moieties is poorly soluble in water due to the bulk lipophilic indole fragments.

The structure of the obtained compounds was characterized by ^1^H and ^13^C NMR, IR spectroscopy and ESI HR mass spectrometry (ESI, Appendix A). Chemical shifts, integrated intensities, and multiplicity of proton signals in ^1^H NMR spectra confirmed the structure of macrocycles **2** and **3**.

It is known that peptide fragments with *L*-Phe and *L*-Trp fragments are prone to the formation of various supramolecular repeating β-structures due to different functional fragments involved in the formation of hydrogen bonds and stabilizing molecules [26,27].

Solid-state IR spectra were examined to evaluate the structural features of the obtained macrocycles **2** and **3**. Considering the IR spectrum (ESI, Appendix A), we observe that for the macrocycle **3** the bands of amide I (~1673 cm^−1^) and amide II (~1536 cm^−1^) are brightly expressed and appear with greater intensity than for macrocycle **2** with *L*-Trp fragments. This fact may indicate a denser packing of macrocycle **3** molecules.

### 2.2. Analysis of Cytotoxicity of **2** and **3** by Colorimetric MTT Test against PC-3 and MCF-7 Tumor Cell Lines

The cytotoxicity of the compounds against tumor cell lines was analyzed using colorimetric MTT test with the detection using plate Infinite M200 microplate reader (Tecan, Switzerland). This test is used to assess the metabolic activity of cells.

According to the results of the study, we found that compounds **2** and **3** exhibit cytotoxic activity against MCF-7 breast adenocarcinoma and PC-3 prostate carcinoma cell lines. Concentrations of semi-maximal inhibition were determined in the range of up to 0.5 µM. The IC_50_ (half maximal inhibitory concentration) values of **2** on PC-3 and MCF-7 was about 0.5 µM in both cell lines (Figure 2). The IC_50_ values of **3** on PC-3 and MCF-7 was about 0.3÷0.4 and 0.5 µM correspondingly. This is an important result in the development of delivery systems in which the pillar[5]arene exhibits additional activity towards tumor cells.

### 2.3. Pillar[5]arene/Fluorescein Nanoparticles

The next stage of the research was to study the association of the synthesized macrocycles with fluorescein (**Fluo**). Fluorescein-containing nanoparticles were formed based on two polycationic compounds—decasubstituted pillar[5]arene derivatives with amino acid moieties of *L*-Tryptophan (**2**) and *L*-Phenylalanine (**3**) on both rims in the macrocyclic structure. The ability of pillar[5]arenes **2**, **3** to interact with **Fluo** was studied by UV–vis, fluorescence spectroscopy, DLS and TEM. Taking into account the solubility of obtained pillar[5]arenes **2** and **3**, ethanol–water solvent system (1:4) was chosen for the study of association with fluorescein.

#### 2.3.1. UV–Vis Spectroscopy

Electron absorption spectroscopy is one of the most illustrative and convenient methods for studying the interactions and evaluating the constants of formation of complexes between organic compounds, including dyes, drugs, and biopolymers and macrocyclic compounds [28].

The interactions of pillar[5]arenes **2**, **3** with **Fluo** were studied by UV–vis spectroscopy. The experiments were carried out in an ethanol–water solvent system (1:4). The solution of **Fluo** was added to the solutions of **2** and **3** (1 × 10^−3^ M) in a 1:1 molar ratio (concentration of components in the mixture was 5 × 10^−5^ M).

Pillar[5]arenes **2** and **3** absorb in the region of 200–300 nm. According to UV–vis spectra, pillar[5]arene **2** has two absorption maxima with λ_max_ at 281 nm and 288 nm in ethanol, the absorption bands due to the excitation of π–π* transitions in the indole group and the macrocyclic backbone of the molecule. Macrocycle **3**-containing *L*-Phenylalanine moieties possess one maximum (λ_max_ = 288 nm) associated with π–electron transitions of the benzene fragments in the amino acid residue and the macrocyclic backbone (ESI, Appendix A) [29].

Earlier it was shown [30] that fluorescein can exist in seven ionization forms such as anion (carboxylate, phenolate), dianion, three neutral forms (zwitterion, *p*–quinoid and lactone), and cation depending on the solution’s pH value (ESI, Appendix A). In ethanol medium, fluorescein is characterized by two maxima at around 450 and 480 nm in the visible part of the absorption spectra. In both pure water and ethanol, it exists predominantly in a monoanionic carboxylate form [30].

The interactions of macrocycle **2** (1 × 10^−5^ M) with **Fluo** was established with the UV–vis spectroscopy (ESI, Appendix A). In this case, the changes in the visible region indicate slight changes in fluorescein microenvironment (pH data of studied solutions ESI, Appendix A) along with intermolecular π–π stacking between the indole fragments of *L*-Tryptophan residues in the pillar[5]arene structure with the dye [31].

However, the interactions of macrocycle **3** (1 × 10^−5^ M) with **Fluo** is indicated by hypochromic effect and insignificant batochromic shift in the visible absorption region (ESI, Appendix A). Considering that stated above, it is likely that the interaction of fluorescein with pillar[5]arene **3** results in the formation of supramolecular associates. Electrostatic interactions between negatively charged parts of the monoanionic carboxylate form of fluorescein and positively charged quaternary nitrogen atoms in the macrocycle structure together with the bunch of other non-covalent interactions also affect the complex formation.

Spectrophotometric titration method was used to determine the association constant. Synthesized macrocycles **2**, **3** do not show absorption within wavelength range 350–550 nm; in this regard, we decided to carry out a spectrophotometric titration with a change in the concentration of macrocycles, while the guest’s concentration remained constant (1 × 10^−4^ M) (the pillar[5]arenes:**Fluo** ratio increased from 0.3:1 to 3:1) (Figure 3).

The results obtained were processed by BindFit [32]. Association constants were calculated assuming 1:1 complex composition. The logarithm of association constant of pillar[5]arene **2** with **Fluo** was found to be lgK_a_(**2**+**Fluo**) = 2.94 (ESI, Appendix A) and lgK_a_(**3**+**Fluo**) = 3.92 (ESI, Appendix A) for macrocycle **3** with **Fluo**. No good fit was obtained when titration data were processed using models with complex compositions of 1:2 and 2:1.

#### 2.3.2. Fluorescence Spectroscopy

Fluorescence spectroscopy is a sensitive method for characterizing interactions in supramolecular systems. The presence of fluorophore fragments (*L*-Phe and *L*-Trp residues) in pillar[n]arenes implies the possibility of having fluorescent properties [33]; indeed, pillar[5]arene **2** fluoresces with a maximum at 350 nm with the excitation wavelength of 290 nm. Fluorescence is due to the emission of tryptophan fragments (ESI, Appendix A). The small Stokes shift (60 nm) indicates the removal of indole fragments from each other in the macrocycle and the absence of excimer formation. Obviously, this is strained with the repulsion of positively charged fragments and their short length, which was previously shown for thiacalix[4]arenes (for them, the Stokes shift reaches 135 nm during the formation of excimers of indole fragments) [34].

The fluorescence of pillar[5]arene **3** is driven by the presence of phenylalanine fragments (ESI, Appendix A). However, it is well known [35,36] that this amino acid residue fluoresces with an emission maximum at fairly short wavelengths. At the same time, fluorescence is observed for the macrocycle **3** with an emission maximum at 320 nm, which is in good agreement with the earlier published data for pillar[5]arenes [37]. Obviously, the fluorescence properties of both macrocycles are conditioned by both the presence of fluorophores and charged groups. Note, similar fluorescent properties have been recently discovered in charged derivatives of thiacalix[4]arene [38], which additionally confirms our assumptions.

The involvement of aromatic fragments of amino acids of the pillar[5]arenes in interactions with the substrate, whether it is the formation of complexes and associates or a change in the polarity of the microenvironment, can change the fluorescence spectrum (intensity, shift of the fluorescence maximum, etc.) [39]. Thus, we recorded the fluorescence spectra of individual substances and their mixtures in the range of macrocycle:**Fluo** ratios from 0.5:1 to 15:1 in ethanol–water solvent system (1:4) (Figure 4a,b), respectively).

Comparing the individual emission spectra of macrocycles **2**, **3**, **Fluo** and their mixture in 1:1 ratio, we can confirm the interaction between pillar[5]arenes and dye by corresponding changes in the spectra. We observe an insignificant bathochromic shift and almost a two-fold quenching of the fluorescein emission for the **2:Fluo**=1:1 mixture (red dashed line) in the wavelength range of 500–550 nm (Figure 4a). On the contrary, for the system **3:Fluo**=1:1 (red dashed line) (Figure 4b), it was observed an increase in emission of the fluorescein by one and a half with a similar bathochromic shift in the same wavelength range. Considering the titration curves of two macrocycles with fluorescein, it is worth noting an important difference: Figure 4b clearly shows that the graph corresponding to the **3:Fluo**=1:1 mixture is strongly out of the spectral picture (red dashed line). This feature can be related to the different emission mechanism for two macrocycles and, correspondingly, the different type of complexes (Figure 5). It is well known that intermolecular π–π stacking is a characteristic type of interaction for most organic dyes with a planar aromatic system. In case of pillar[5]arenes **2** and **3**, planar aromatic systems are a macrocyclic aromatic platform and its aromatic substituents. Obviously, fluorescence quenching in the case of **2** is associated with the intermolecular interaction of fluorescein with the indole system of substituents and incorporation between macrocycle molecules (Figure 5). Moreover, a common emission mechanism for these systems is aggregation-caused quenching (ACQ) [40,41].

Turning to the features of the spectral pattern for the **3**+**Fluo** system (Figure 4b), we observe a slight emission of **Fluo** within the range of 500–550 nm. The spectrum of **3**+**Fluo** (1:1) clearly shows the spontaneous intensification of the emission and its further quenching at macrocycle:**Fluo** ratios other than equimolar. Based on this fact, it can be assumed that at an equimolar ratio of reagents, the initial components are aggregated with the formation of a stable associates, where **Fluo** incorporate between pillar[5]arene molecules. With an increase in the dye concentration, quenching is observed. It is associated with the blocking of substituents due to π–π stacking. This system correlates well with the relatively new concept of aggregation-induced emission (AIE) [42].

#### 2.3.3. 2D NOESY NMR Spectroscopy of Pillar[5]arene–Fluorescein Mixtures

To confirm our assumptions about the structure of associates **2**+**Fluo** and **3**+**Fluo** and the types of intermolecular interactions between macrocycles and fluorescein molecules that affect the quenching mechanism, the 2D ^1^H–^1^H NOESY NMR method was used. Solutions of the compounds **2** and **3** were prepared at 1 × 10^−2^ M, complexation with **Fluo** (1 × 10^−2^ M) was studied at a 1:1 ratio.

The spectrum of system **2**+**Fluo** (Figure 6a) shows cross-peaks between the aromatic H^i^ protons of **Fluo** and the H^17^ protons of the indole fragment of the macrocycle **2**. It confirms the assumption about convergence of planar aromatic fragments of the tryptophan fragment in the case of pillar[5]arene **2** and fluorescein by π–π stacking (Figure 6b and Appendix A). Noteworthy are the number of correlations between the aromatic H^c^, H^d^ protons of **Fluo** and methyl H^7^ protons at the quaternary nitrogen atom and H^10^ protons in pillar[5]arene **2** structure. This allows us to state about the contacts of oppositely charged fragments of two molecules. Thus, it can be assumed that the dye molecules cooperate with macrocycle **2**, surrounding the pillar[5]arene core, through weak non-covalent interactions.

A different picture is observed for the system **3**+**Fluo** (ESI, Appendix A). The two-dimensional spectrum of the **3/Fluo** system is characterized by the presence of a larger number of cross-peaks between the aromatic protons of the dye and the aliphatic protons of the macrocycle **3** (ESI, Appendix A) comparing to the system **2/Fluo**. In this case, we can assume a significant role of electrostatic interactions and hydrogen forces in the binding of fluorescein with pillar[5]arene **3**. This allows the dye molecules to be integrated between macrocycles and retained there due to electrostatic interactions (-COO^−^ and quaternary N^+^ fragments). At the same time, the terminal phenyl groups of **3** remain free, which leads to an increase in fluorescence (Figure 5).

2D ^1^H–^1^H NOESY NMR spectra are in good agreement with the fluorescence data and confirm the proposed structures of associates that affect the manifested fluorescence mechanisms (Figure 5).

#### 2.3.4. Computer Simulation Studies

In order to clarify the nature of intermolecular interactions, we have involved computer simulation studies. We performed preliminary molecular dynamics simulations which have shown that the substituents of the studied pillar[5]arenes are very flexible and adopt many different conformations within a short simulation time. This fact is in good agreement with the data obtained in the study of the fluorescent properties of macrocycles on the absence of convergence of indole fragments [34,37]. The interactions between fluorescein and pillar[5]arenes do not lead to formation of a complex with a rigid structure. In simulations, the fluorescein molecule easily moves around the pillararene ring interacting non-covalently with different substituents and sampling a huge number of different poses in the absence of a deep global energy minimum.

In Figure 7 and Appendix A (ESI) three top scoring poses for each pillar[5]arene **2** are shown. In the energy minima, the fluorescein molecule is always located between several substituents of the macrocyclic ring, but can have an arbitrary position and orientation with respect to the pillararene ring itself. The intermolecular interactions stabilizing pillararene–fluorescein binding include hydrogen bonding, electrostatic attraction between oppositely charged atoms, hydrophobic interactions, and π–stacking interactions. We found that fluorescein participates in slightly more hydrogen bonds with pillar[5]arene **3** containing phenyl groups (1.65 bonds in average versus 1.35 bonds for pillar[5]arene **2**) (ESI, Appendix A). In contrast, more π–stacking interactions with fluorescein are observed in the case of macrocycle **2** containing a large aromatic tryptophan group (Figure 6a). As mentioned above, the particular poses are not the stable structures. The contacts between fluorescein and any of the pillararene substituents can be lost after just a hundred of picoseconds of molecular dynamics simulation at room temperature.

We can conclude that the location of the fluorescein molecule in complexes plays a key role in the formation of systems with different behavior. In the case of macrocycle **2**, as shown above, π–π stacking interactions predominate, as a result of which aggregation-induced quenching is observed. Turning to macrocycle **3**, hydrogen bonding predominates, which leads to the complex formation of another type, as a result of which the fluorescence buildup is observed at an equimolar pillar[5]arene–**Fluo** ratio.

Thus, the methods used (fluorescence, NMR and computer simulation studies) to establish the structure of associates are in full agreement with each other and unambiguously confirm our assumptions about the structure of macrocycle associates with fluorescein (Figure 5).

#### 2.3.5. Self-Assembly of Pillar[5]arene Derivatives **2**, **3** and Fluorescein

Fluorescence studies show two mechanisms for **2**+**Fluo** and **3**+**Fluo**. Both of these mechanisms are caused by aggregation. Therefore, we applied the dynamic light scattering (DLS) method to study the ability of macrocycles **2** and **3** to self-assemble nanosized associates. This method allows to determine the hydrodynamic diameter of the self-aggregates, as well as their stability and charge [43]. The concentrations of pillar[5]arenes **2** and **3** were ranged from 1 × 10^–4^ to 1 × 10^–6^ M (ESI, Appendix A). The hydrodynamic particle size and polydispersity of the systems were measured at 20 °C.

In the range of studied concentrations for both macrocycles **2** and **3,** the formation of an unstable polydisperse system was observed with the exception of colloidal solutions with high concentrations (ESI, Appendix A). Macrocycle **2** at 1 × 10^–4^ M is characterized by a polydispersity index within 0.36÷0.04 with particle size 117÷19 nm (ESI, Appendix A) (ESI, Appendix A). At the same conditions, pillar[5]arene **3** (1 × 10^–4^ M) forms particles with sizes around 223÷22 nm in a solution (PDI = 0.34 ÷ 0.04) (ESI, Appendix A). Further dilution of the initial concentrated solution of macrocycle **3**-containing *L*-Phenylalanine residue led to an increase in the polydispersity of the system. In the case of macrocycle **2**, a similar trend was observed; the polydispersity indices of dilute solutions were too high.

Before describing the particles, it is worth mentioning the hydrodynamic characteristics of individual **Fluo** in ethanol solution; the hydrodynamic diameter of particles formed by fluorescein was about 169 ÷ 55 nm, PDI = 0.40 ÷ 0.01 (ζ = −5 ÷ 2 mV) (ESI, Appendix A).

#### 2.3.6. Aggregation Study of Pillar[5]arenes **2** and **3** with **Fluo** by DLS and TEM

The formation of nano-sized aggregates was observed for both macrocycles **2** and **3** with fluorescein in 1:1 molar ratio in ethanol–water solvent system (1:4). According to DLS data, the average hydrodynamic diameter of particles formed by macrocycle **2** containing *L*-Trp moieties with **Fluo** was 92÷1 nm (PDI = 0.25 ÷ 0.01) (ESI, Appendix A). pillar[5]arene **3**+**Fluo** (1:1) nanoparticles turned out to be more compact (d = 79 ÷ 1 nm) and the dispersion of the solution for the system **3**+**Fluo** was also lower than for the **2**+**Fluo** (PDI = 0.16 ÷ 0.01) (ESI, Appendix A).

Zeta-potential (ζ) is an additional indicator that characterizes the stability of a colloidal system. According to the literature data, the colloidal system is stable and not prone to aggregation and further sedimentation of particles in it, if the zeta-potential value is 30 mV or more modulo [44]. For the system **2**+**Fluo,** the value of zeta-potential is 13÷2 mV, which characterizes this colloidal system as unstable. On the contrary, the colloidal system of nanoparticles **3**+**Fluo** turned out to be more stable (ζ = 32 ÷ 2 mV).

The presence of nano-sized particles **2**+**Fluo** and **3**+**Fluo** was also confirmed by TEM (Figure 8). TEM images clearly show the presence of spherical particles for both systems (**2**+**Fluo** and **3**+**Fluo**). However, in the case of the system **Fluo**–pillar[5]arene **2** (ESI, Appendix A) the formation of clumped particles with fuzzy boundaries was observed (Figure 8a). At the same time, macrocycle **3** with **Fluo** form isolated nanoparticles with clear boundaries (Figure 8b) (ESI, Appendix A).

The data obtained by DLS and TEM are in good agreement with the assumptions put forward in the description of the UV–vis spectra and fluorescent spectroscopy, which additionally were confirmed by computer simulation studies. We pointed out the demonstration of different types of interactions during the formation of nanoparticles based on pillar[5]arenes **2** with *L*-Tryptophan, **3** with *L*-Phenylalanine moieties, and fluorescein.

Additionally, DLS and TEM data are consistent with the data of fluorescence spectroscopy. The assumption about different emission mechanisms was confirmed (ACQ for **2**+**Fluo** and AIE–**3**+**Fluo**). For particles based on fluorescein and pillar[5]arene **3** with *L*-Phenylalanine fragments, we observed the formation of a more stable system of nanoparticles in which fluorescein may be incorporated into pseudocavities of macrocycle **3** self-associates. In such particles, it turns out to be logical that the entropy of the guest molecule decreases due to restriction of intramolecular motions in the aggregate state, as the main working AIE mechanism [45].

#### 2.3.7. Study of the Fluorescein Release from Pillar[5]arene/Fluorescein Nanoparticles at Different pH

The reversibility of the interaction between reactive components in supramolecular systems is an important condition for their application [46]. Sensitive systems with the controlled release of drugs or diagnostic markers are attractive for solving the problems of biomedicine and antitumor therapy. In connection with this, fabrication and development of bio and chemosensors based on the fluorescein and new type of macrocycle compounds pillar[5]arenes with the study of release of bound substrate are still of tremendous interest.

The following step of the work was the study of the possibility of fluorescein releasing from equimolar complexes **2**+**Fluo** and **3**+**Fluo** (C = 1 × 10^−5^ M) under the change in the acidity of the solution. Thus, solutions of the binary system were prepared in acetate buffer (pH = 4.5), phosphate buffer (pH = 7.4), and sodium tetraborate buffer (pH = 9.2). It is known that the visible region in the absorption spectra for the fluorescein solution is sensitive to changes in the acidity of the medium and microenvironment; that is why the fluorescein release from the systems was observed within 400–550 nm in the visible region of the UV–vis spectra.

A considerable increase in the absorption intensity of fluorescein at about 490 nm (Figure 9a) occurs at 30 min after mixing in the case of dilution of an aliquot of the binary system with a phosphate buffer solution (pH = 7.4). A gradual increase in absorption intensity continued during the first two hours, indicating the release of fluorescein. However, we observe a decrease in the intensity in the visible region of the spectrum a day after the preparation of the solution. This turned out to be even lower than the absorption intensity of the freshly prepared mixture. In this case, the system demonstrated a time-controlled release of the dye. The additional fact of dye release and destruction or rearrangement of complexes is the formation of the polydisperse system, PDI = 0.75 ÷ 0.10 (ESI, Appendix A, Appendix A).

In the case of changing the pH medium to a more alkaline one (sodium tetraborate buffer, pH = 9.2) (Figure 9b), we observed an increase in the absorption intensity with a slight hypsochromic shift along with a rise in the baseline throughout the study period. These changes made it possible to establish the release of fluorescein from the equimolar associate **2**+**Fluo**. However, in this case, a bimodal system was formed (PDI = 0.41 ÷ 0.03) (ESI, Appendix A, Appendix A). It should be noted that the formation of turbidity was observed in both cases for the **2**+**Fluo** system in neutral (pH = 7.4) and alkaline (pH = 9.2) buffers during the study period, followed by precipitation.

In the case of the **2**+**Fluo** (pH = 4.5) and **3**+**Fluo** systems (in all studied buffer solutions), no significant changes were observed in the absorption spectra and polydispersity indices of the systems (ESI, Appendix A, Appendix A), with the exception of the **3**+**Fluo** system in phosphate buffer (ESI, Appendix A, Appendix A), for which a significant enlargement of particles was recorded (d = 802 ÷ 18 nm).

Thus, the release of fluorescein from the equimolar **2**+**Fluo** associates over time in neutral (pH = 7.4) and alkaline (pH = 9.2) solutions was shown by electron absorption spectroscopy and DLS. On the other hand, there is no release of the dye from the **2**+**Fluo** associate at pH = 4.5. For the **3**+**Fluo** associate, there was also no release observed in all media studied, which is more logically consistent with the stability of the initial colloidal system **3**+**Fluo** (ζ = 32 ÷ 2 mV) and the greater value of the association constant (lgK_a_(**3**+**Fluo**) = 3.92), than for the system **2**+**Fluo**.

## 3. Methods and Materials

### 3.1. General

The ^1^H, ^13^C (^13^C{^1^H}—100 MHz and ^1^H—400 MHz) and 2D ^1^H–^1^H NOESY (400.0 MHz) NMR spectra were obtained on a Bruker Avance–400 spectrometer (Bruker Corp., Billerica, MA, USA). The chemical shifts were determined against the signals of residual protons of deuterated solvent (D_2_O, DMSO–d_6_, CD_3_OD–d_4_). The concentrations of the compounds were equal to 3–5% by the weight in all the records. Spectrum 400 (Perkin Elmer Inc., Waltham, MA, USA) IR spectrometer, Agilent 6550 iFunnel Q–TOF LC/MS (Agilent Technologies, Santa Clara, CA, USA), equipped with Agilent 1290 Infinity II LC, were applied for the IR and ESI HRMS recording, respectively, and PerkinElmer 2400 Series II for elemental analysis. Melting points were determined using Boetius Block apparatus (VEB Kombinat Nagema, Radebeul, Germany). DataAnalysis 4.0 software (Bruker Daltonik GmbH, Bremen, Germany) was used for spectra analysis. Organic solvents were purified in accordance with standard procedures. All the aqueous solutions were prepared with the Millipore-Q deionized water (>18.0 MW cm at 25 ºC).

### 3.2. Synthesis

Alkylating agents based on L-tryptophan and L-phenylalanine were prepared by a literature method [22].

**4,8,14,18,23,26,28,31,32,35-Decakis-[(N-30,30-dimethylaminoethyl)carbamoyl methoxy]-pillar**[5]**arene (1)** was prepared by a literature method [23].

#### General Procedure for the Synthesis of Compounds **2**, **3**

In a round-bottom flask equipped with magnetic stirrer 0.2 g (0.112 mmol) of decaamine **1** was dissolved in 5 mL of acetonitrile, and 1.232 mmol of alkylating reagent were added. The reaction mixture was refluxed for 12–14 h. Solvent was removed under reduced pressure. The product was recrystallized from propanol–2. The precipitates were dried under reduced pressure over P_2_O_5_.

**4,8,14,18,23,26,28,31,32,35-Decakis-[(N-[2-dimethyl({ethoxycarbonyl[S-indole-3-yl-methyl]methyl}aminocarbonylmethyl)ammonio]ethyl)aminocarbonylmethoxy]-pillar[5]arene decabromide** (**2**). Yield: 0.61 g (82%). M.P. = 165–171 °C.

^1^H NMR (DMSO-d_6_, 400 MHz, 298 K): δ, ppm, J/Hz: 1.08 (t, 30H, CH_3_CH_2_O,^3^J_HH_ = 7.2), 2.97–3.19 (m, 40H, NHCHCH_2_, NHCH_2_CH_2_N^+^), 3.22 (s, 60H, N^+^(CH_3_)_2_), 3.61–3.80 (m, 30H, ArCH_2_Ar, NHCH_2_CH_2_N^+^), 3.96–4.07 (m, 20H, CH_3_CH_2_O), 4.24–4.37 (m, 10H, NHCHC(O)), 4.38–4.52 (m, 20H, ArOCH_2_), 4.53–4.61 (m, 20H, N^+^CH_2_C(O)), 6.85 (s, 10H, ArH^pillar^), 6.99 (t, 10H, ArH^indol^, ^3^J_HH_ = 7.3), 7.07 (t, 10H, ArH^indol^, ^3^J_HH_ = 7.5), 7.15 (d, 10H, NHCH^indol^, ^3^J_HH_ = 7.3), 7.34 (t, 10H, ArH^indol^, ^3^J_HH_ = 7.1), 7.48 (t, 10H, ArH^indol^, ^3^J_HH_ = 7.7), 8.45–8.51 (m, 10H, ArOCH_2_C(O)NH), 9.19 (br.t, 10H, C(O)NHCH), 10.93 (s, 10H, NH^indol^).

^13^C NMR (DMSO-d_6_, 100 MHz, 298 K) δ, ppm: 13.9, 25.5, 26.8, 51.3, 53.5, 60.9, 62.0, 63.5, 108.7, 111.5, 117.9, 118.5, 121.1, 124.1, 126.9, 136.1, 163.2, 170.9.

IR (ν/cm^−1^): 1202 (C–O–C), 1526 (N–H), 1670 (C=O), 1733 (COO–Et), 3201 (N–H).

MS (ESI): calculated [M − 9Br^−^, -4C_2_H_5_^•^, +4H^+^]^13+^m/z = 353.3223, [M − 10Br^−^, -10C_2_H_5_^•^, -4C_14_H_16_NO_2_^•^, +6H^+^]^16+^m/z = 224.9903, found [M − 9Br^−^, -C_2_H_5_^•^, +4H^+^]^13+^m/z = 353.0494, [M − 10Br^−^, -10C_2_H_5_^•^, -4C_14_H_16_NO_2_^•^, +6H^+^]^16+^m/z = 224.1293.

El. anal. calcd for C_245_H_320_Br_10_N_40_O_50_: C 54.25; H 5.95; Br 14.73; N 10.33. Found: C 53.91; H 6.02; Br 14.11; N 10.14.

**4,8,14,18,23,26,28,31,32,35-Decakis-[(N-[2-dimethyl({ethoxycarbonyl[S-benzyl]methyl} aminocarbonylmethyl)ammonio]ethyl)aminocarbonylmethoxy]-pillar**[5]**arene decabromide** (**3**): Yield: 0.56 g (80%). M.P. = 133–137 °C.

^1^H NMR (DMSO–d_6_, 400 MHz, 298 K): δ, ppm, J/Hz: 1.11 (t, 30H, CH_3_CH_2_O, ^3^J_HH_ = 7.2), 2.88–3.01 (m, 20H, CHCH_2_Ph), 3.02–3.13 (m, 20H, NHCH_2_CH_2_N^+^), 3.20 (s, 60H, N^+^(CH_3_)_2_), 3.53–3.81 (m, 30H, NHCH_2_CH_2_N^+^, ArCH_2_Ar), 4.05 (q, 20H, CH_3_CH_2_O, ^3^J_HH_ = 7.0), 4.19–4.27 (m, 10H, CHCH_2_Ph), 4.39–4.48 (m, 20H, ArOCH_2_), 4.51–4.62 (m, 20H, N^+^CH_2_C(O)), 6.85 (s, 10H, ArH), 7.18–7.33 (m, 50H, PhH), 8.45–8.50 (m, 10H, ArOCH_2_C(O)NH), 9.19–9.25 (m, 10H, C(O)NHCH).

^13^C NMR (DMSO-d_6_, 100 MHz, 298 K) δ, ppm: 13.9, 28.8, 36.6, 51.3, 60.7, 60.9, 62.8, 126.8, 128.4, 129.2, 136.6, 163.2, 168.8, 170.6.

IR (ν/cm^−1^): 1202 (C–O–C), 1527 (N–H), 1673 (C=O), 1734 (COO–Et), 3193 (N–H).

MS (ESI): calculated [M − 7Br^−^, -7C_2_H_5_^•^, -C_11_H_13_O_2_^•^, +H^+^]^8+^m/z = 512.2149, [M − 10Br^−^, -4C_2_H_5_^•^, -2C_11_H_13_O_2_^•^, +2H^+^]^12+^m/z = 314.1650, found [M − 7Br^−^, -7C_2_H_5_^•^, -C_11_H_13_O_2_^•^, +H^+^]^8+^m/z = 512.2760, [M − 10Br^−^, -4C_2_H_5_^•^, -2C_11_H_13_O_2_^•^, +2H^+^]^12+^m/z = 314.0389.

El. anal. calcd for C_225_H_310_Br_10_N_30_O_50_: C 53.68; H 6.21; Br 15.87; N 8.35. Found: C 51.92; H 6.56; Br 16.1; N 7.84.

### 3.3. UV–Visible Spectroscopy

Absorption spectra were recorded on a Shimadzu UV–3600 spectrometer (Kyoto, Japan). Quartz cuvettes with an optical path length of 10 mm at 25 °C were used. Ethanol was used for preparation of the **2** and fluorescein solutions, water was used for preparation of the **3** solution. Absorption spectra of mixtures were recorded after an 1 h incubation at 25 °C. Solution of the fluorescein was added to those of compounds **2**, **3** (both in a 1 × 10^−3^ M) in a 1:1 molar ratio to study the association of pillar[5]arenes with guest in ethanol–water solution (concentration of components is 1 × 10^−5^ M).

### 3.4. Determination of the Association Constant by Spectrophotometric Titration

A 150 µL of a fluorescein solution (2 × 10^−3^ M) in ethanol was added to 1 × 10^−3^ M solution of **2** and **3** (90, 150, 210, 270, 300, 360, 450, 540, 600,750 and 900 µL) in ethanol and water respectively and diluted to final volume of 3 mL with water. The volume of ethanol was constant for each system and ratio, and was 25% by volume. The UV–vis spectra of the solutions were then recorded. The association constant was calculated by BindFit. Three independent experiments were carried out for each series.

### 3.5. Fluorescence Spectroscopy

Fluorescence spectra were recorded on the Fluorolog 3 luminescent spectrometer (Horiba Jobin Yvon, Longjumeau, France). For the study of association of pillar[5]arenes **2**, **3** with fluorescein various macrocycle/guest ratios were chosen. The excitation wavelength was selected as 290 nm, the emission scan range was 300–565 nm and excitation and emission slits were 2 nm. Quartz cuvettes with an optical path length of 10 mm were used. Cuvette was placed at the front face position to avoid the inner filter effect. Fluorescence spectra were automatically corrected by the Fluorescence program. Spectra were recorded at 25 °C in ethanol–water system. A 300 µL of a fluorescein solution (1 × 10^−4^ M) in ethanol was added to 1 × 10^−3^ M solution of **2** (15, 30, 90, 210, 300 and 450 µL) in ethanol or a 1 × 10^−3^ M solution of **3** (15, 30, 90, 210, 300 and 450 µL) in water and diluted to final volume of 3 mL with water.

### 3.6. The Study of Fluorescein Release from the Associates with Pillar[5]arenes **2**, **3** over Time by UV–Visible Spectroscopy

Quartz cuvettes with an optical path length of 10 mm at 25 °C were used. Spectra were recorded at 25 °C in ethanol–water system. A 1 × 10^−3^ M solution of **2** (30 µL) in ethanol and a 1 × 10^−3^ M solution of **3** (30 µL) in water respectively were alternately added to 300 µL of a fluorescein solution (1 × 10^−4^ M) in ethanol and diluted in series to final volume of 3 mL with 0.1 M acetate buffer (pH = 4.5), 0.01 M phosphate buffer saline (PBS, Sigma, 0.0027 KCl, 0.137 NaCl, pH 7.4), 0.01 M sodium tetraborate buffer (pH = 9.2). The change in the absorption of the resulting solutions was observed over time: immediately, after 30 min, 1 h, 2 h, and one day after the preparation of the solutions. At the same condition absorption spectra were recorded for an individual solution of fluorescein.

### 3.7. Dynamic Light Scattering (DLS)

Particles size. The particles size was determined by the Zetasizer Nano ZS instrument at 25 °C. The instrument contains 4 mW He–Ne laser operating at a wavelength of 633 nm and incorporated noninvasive backscatter optics (NIBS). The measurements were performed at the detection angle of 173° and the software automatically determined the measurement position within the quartz cuvette. The experiments were carried out for each solution in triplicate. Synthesized pillar[5]arenes **2**, **3** were dissolved completely in ethanol and water respectively at concentrations used in research. For the determination pillar[5]arene:fluorescein associates size in water samples were prepared with equilibrium concentration 1 × 10^−4^ M.

For the determination pillar[5]arene:fluorescein associates size in water and in different buffers samples were prepared in the same way as UV–vis measurement.

### 3.8. Zeta Potentials

Zeta potentials were measured on a Zetasizer Nano ZS from Malvern Instruments. Samples were Prepared as for the DLS measurements and were transferred with a syringe to a disposable folded capillary cell for measurement. The zeta potentials were measured using the Malvern M3–PALS method, and the reported values were taken from the average of three measurements.

### 3.9. Transmission Electron Microscopy (TEM)

TEM analysis of nanoparticles based on synthesized macrocycles **2**, **3** and fluorescein was carried out using the Hitachi HT7700 Exalens transmission electron microscope with Oxford Instruments X–Max^n^ 80T EDS detector. For sample preparation, 10 μL of the suspension were placed on the Formvar™/carbon coated 3 mm copper grid, which was then dried at room temperature. After complete drying, the grid was placed into the transmission electron microscope using special holder for microanalysis. Analysis was held at the accelerating voltage of 80 kV in STEM mode using Oxford Instruments X–Max^n^ 80T EDS detector.

### 3.10. Computer Simulation Studies

All computational studies were performed using OPLS3e force field in the Schrödinger Suite software.

In order to characterize favorable contacts between fluorescein and pillar[5]arenes at the local energy minima, we docked a fluorescein molecule to the different static conformations of pillar[5]arenes obtained by simulated annealing technique. For the conformer generation, each pillar[5]arene molecule was solvated in water containing 0.15 M NaCl and neutralized by additional chloride counter–ions. The system was equilibrated in NPT ensemble at 298 K and 1 bar pressure and then quickly heated to 1000 K at the constant volume and kept at this temperature during 150 ps. Then the system was cooled down to 298 K, equilibrated for 700 ps and cooled down to 50 K to obtain the conformer corresponding to the local energy minimum. This procedure was repeated 20 times for each pillar[5]arene. After that, a fluorescein molecule in the carboxylate monoanionic form which is prevalent in its aqueous solution was blindly docked to each of 20 conformers using Glide XP scoring function. The conformers were kept rigid while fluorescein was flexible.

### 3.11. Cytotoxicity Analysis of Compounds ***2*** and ***3*** by Colorimetric MTT-Test against PC-3 and MCF-7 Tumor Cell Lines

#### 3.11.1. Cell Culture Cultivation

Tumor cell lines MCF-7 (breast adenocarcinoma) and PC-3 (prostate carcinoma) were cultured in DMEM media with the addition of 5% inactivated fetal bovine serum, 1 mM L–glutamine, and a mixture of penicillin (5000 U/mL) and streptomycin (5000 ug/mL). The cells were cultured at 37 °C in a humid atmosphere containing 5% CO_2_. The cells were placed in 96-well culture plates with a flat bottom at a concentration of 4 × 10^3^ cells per well in a full culture medium and incubated at 37 °C with 5% CO_2_ for 24 h.

#### 3.11.2. Addition of the Studied Compounds to Cells

Stock solutions of the compounds were prepared before the experiment by dissolving in DMSO to a final concentration of 20 mM. The studied compounds were added to the wells with cells to obtain the final concentration of 0.5 µM, 1 µM, 1.5 µM, 2.5 µM, 5 µM, 10 µM, 20 µM. The final concentration of DMSO in each well was 1%. As a vehicle control, DMSO was added to untreated cells until the final concentration of 1%.

#### 3.11.3. Cytotoxicity Analysis of the Compounds

The cytotoxicity of the compounds against tumor cell lines was analyzed using colorimetric MTT-test with the detection using plate Infinite M200 microplate reader (Tecan, Switzerland). This test is used to assess the metabolic activity of cells. 48 h after the addition of compounds, the medium was removed from each well and 100 µL of fresh medium with MTT reagent was added until the final MTT concentration of 0.5 mg/mL. Then the plates were placed in a CO_2_ incubator at 37 °C for 4 h. After that, 150 mL of dimethyl sulfoxide (DMSO) solution was added to the wells and incubated for another 15 min on a shaker. The optical density was measured at a wavelength of 590 nm using Infinite M200 microplate reader (Tecan, Männedorf, Switzerland). Cell viability was calculated as a relative value, taking the optical density values in the control wells as 100%. Data processing was carried out in the GraphPad Prism 8.

## 4. Conclusions

We have successfully synthesized new decasubstituted pillar[5]arene derivatives containing *L*-Tryptophan and *L*-Phenylalanine residues and showed the ability of the synthesized macrocycles to form nanoparticles with fluorescein. The morphology of the resulting particles depends on the amino acid residue in the pillar[5]arene structure. It was shown that macrocycle with *L*-Phenylalanine fragments binds the dye more efficiently (lgK_a_ = 3.92). The obtained nanoparticles based on the synthesized macrocycles and fluorescein in an equimolar ratio had a different fluorescence ability, which can be explained by different mechanisms such as ACQ for the **2**+**Fluo** and AIE–**3**+**Fluo**. It was found that the dye release occurs in the neutral (pH = 7.4) and alkaline (pH = 9.2) buffer solutions only for the associate based on a macrocycle containing *L*-Tryptophan fragments, while pillar[5]arene **3/Fluo** nanoparticles remain stable in all the studied conditions. The cytotoxic effect of pillar[5]arenes **2** and **3** on tumor cell lines in the MTT test was characterized. The compounds showed a statistically significant cytotoxic activity against MCF-7 breast adenocarcinoma and PC-3 prostate carcinoma cell lines. Thus, pillar[5]arene/**Fluo** nanoparticles perform a dual function: therapeutic and dosing. They are able to suppress tumor cells and at the same time act as a container for a dosing regimens control. The results can be used to create new anticancer materials as dosing carrier with controlled substrate release.

## Data Availability

The data presented in this study are available in Appendix A.

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
