# Peer review of "Decasubstituted Pillar[5]arene Derivatives Containing L-Tryptophan and L-Phenylalanine Residues: Non-Covalent Binding and Release of Fluorescein from Nanoparticles"

_ijms, 2023, doi:10.3390/ijms24097700_

Round 1
Reviewer 1 Report
In this paper the authors synthesized new decasubstituted pillar[5]arene derivatives containing L‒tryptophan and L‒phenylalanine residues that form nanoparticles with fluorescein. In addition, they reached numerous significant results: macrocycle with L‒phenylalanine fragments binds the dye more efficiently, the obtained nanoparticles based on the synthesized macrocycles and fluorescein in an equimolar ratio had different fluorescence ability, dye release occurs in the neutral (pH=7.4) and alkaline (pH=9.2) buffer solutions only for the associate based on a macrocycle containing L‒tryptophan fragments, the examined compounds showed a statistically significant cytotoxic activity against MCF-7 breast adenocarcinoma and PC-3 prostate carcinoma cell lines.
The authors wrote: „Thus, pillar[5]arene/Fluo nanoparticles performs a dual function: therapeutic and diagnostic. They are able to suppress tumor cells and at the same time act as a container for a dianostic marker (mising g). The results can be used to create new anticancer materials with sumultanious diagnostic property with controlled substrate release“.
I think this should be corrected since diagnostic marker, and diagnostic property are wrong terms in this case, you could say a dosing carrier, because their role is not to make a diagnosis.
The research is interestingly designed, the paper is excellently written, and the results are significant. However, I find too many self-citations by authors which is wrong. This should be corrected.
Author Response
Dear Reviewer,
Thank you very much for careful consideration of the manuscript. The following modifications were made in accordance with your comments.
In this paper the authors synthesized new decasubstituted pillar[5]arene derivatives containing L‒tryptophan and L‒phenylalanine residues that form nanoparticles with fluorescein. In addition, they reached numerous significant results: macrocycle with L‒phenylalanine fragments binds the dye more efficiently, the obtained nanoparticles based on the synthesized macrocycles and fluorescein in an equimolar ratio had different fluorescence ability, dye release occurs in the neutral (pH=7.4) and alkaline (pH=9.2) buffer solutions only for the associate based on a macrocycle containing L‒tryptophan fragments, the examined compounds showed a statistically significant cytotoxic activity against MCF-7 breast adenocarcinoma and PC-3 prostate carcinoma cell lines.
The authors wrote: „Thus, pillar[5]arene/Fluo nanoparticles performs a dual function: therapeutic and diagnostic. They are able to suppress tumor cells and at the same time act as a container for a dianostic marker (mising g). The results can be used to create new anticancer materials with sumultanious diagnostic property with controlled substrate release“.
I think this should be corrected since diagnostic marker, and diagnostic property are wrong terms in this case, you could say a dosing carrier, because their role is not to make a diagnosis.
Answer:
We corrected Conclusions.
“Thus, pillar[5]arene/Fluo nanoparticles performs a dual function: therapeutic and dosing. They are able to suppress tumor cells and at the same time act as a container for a dosing regimens control. The results can be used to create new anticancer materials as dosing carrier with controlled substrate release. ”
The research is interestingly designed, the paper is excellently written, and the results are significant. However, I find too many self-citations by authors which is wrong. This should be corrected.
Answer:
We corrected references and exchanged some our self-citation by other articles [25, 37].

Reviewer 2 Report
The authors synthesized pillar[5]arene derivatives with amino acids tryptophan and phenylalanine, and examined the association of these derivatives with Fluorescein. I consider these researches to be very interesting and important in the study of host (pillar[5]arene derivatives) and guest (Fluorescein) type molecular complexes. Therefore, it is very important that this manuscript be published after the necessary corrections.
Major:
The main drawback of the manuscript is the absence of a description of the experimental methods, as well as the materials. Authors in the revised paper must provide a description (or link to previously published titles) of each experimental method as well as the recipe for the synthesis of derivatives 2 and 3. Authors must state the origin and purity of each chemical.
Minor:
1. In the introduction, the authors could provide a figure showing the basic (initial model) of pillar[5]arene (both enantiomers) and describe the planar chirality in 1-2 sentences, they could also give the structure of Fluorescein on this figure.
2. In the UV-VIS experiment, the authors concluded that derivative 2 and 3 with Fluorescein forms a 1:1 stoichiometry complex, and they obtained the association constant using the BindFit method. Usually, for 1:1 complexes, the association constant is obtained by the Benesi–Hildebrand method. The authors could highlight the advantage of the BindFit method compared to the Benesi–Hildebrand method, does the association constant differ if calculated by the Benesi–Hildebrand method?
5. The authors could explain in a little more detail (structurally or stereochemically) why Fluorescein in derivative 2 prefers to bind to the indole system and in derivative 3 it prefers to enter between the pillar[5]arene systems. Is there a 2D NOESY NMR cross-peak that corresponds to the interaction of Fluorescein and benzene hydrogens from the pillar[5]arene skeletons, since in Figure 4 for derivative 2 one molecule of fluorescein is also located between the pillar[5]arene skeleton. From the 2D NOESY NMR spectrum, it seems that only the phenylcarboxylic group of Fluorescein interacts with the indole system, while the tricyclic system interacts with the quaternary amine group of derivative 2.
6. The authors could present how the structure of Fluorescein changes depending on the pH.
7. The authors could discuss in 1-2 sentences what would be expected with Fluorescein if the base skeleton of pillar[n]arene n were greater than 5, say 10 or 15.
Author Response
Dear Reviewer,
Thank you very much for careful consideration of the manuscript. The following modifications were made in accordance with your comments.
The authors synthesized pillar[5]arene derivatives with amino acids tryptophan and phenylalanine, and examined the association of these derivatives with Fluorescein. I consider these researches to be very interesting and important in the study of host (pillar[5]arene derivatives) and guest (Fluorescein) type molecular complexes. Therefore, it is very important that this manuscript be published after the necessary corrections.
Major:
The main drawback of the manuscript is the absence of a description of the experimental methods, as well as the materials. Authors in the revised paper must provide a description (or link to previously published titles) of each experimental method as well as the recipe for the synthesis of derivatives 2 and 3. Authors must state the origin and purity of each chemical.
Answer:
Thank you for comment. We have updated the article in accordance with your comments and moved Experimental section from Supporting Information into main article.
Minor:
- In the introduction, the authors could provide a figure showing the basic (initial model) of pillar[5]arene (both enantiomers) and describe the planar chirality in 1-2 sentences, they could also give the structure of Fluorescein on this figure.
Answer:
Figure showing the initial model of pillar[5]arene (both enantiomers) was added in the manuscript. The structure of the fluorescein is given on Figure 5.
We added the following sentence in the introduction “Their distinctive feature is the presence of planar chirality (the chirality plane) (Fig. 1).”
And the following text in the synthetic part of the manuscript “Pillararenes 2 and 3 have asymmetric carbon atoms in the phenylalanine and tryptophan fragments. The reaction can proceed stereospecifically with the predominance of one of the conformers when the initial decamine 1 interacts with alkylating agents with chiral centers. However, earlier our research group established that in this case the reaction products exist not as racemic mixtures, but as a mixture of two conformers with the predominance of one of them [23].”
Based on our earlier investigations [10.3390/ijms21197206] we suppose that in this case pillar[5]arenes 2 and 3 also have to exist as a mixture of two conformers with the predominance of one of them.
- In the UV-VIS experiment, the authors concluded that derivative 2 and 3 with Fluorescein forms a 1:1 stoichiometry complex, and they obtained the association constant using the BindFit method. Usually, for 1:1 complexes, the association constant is obtained by the Benesi–Hildebrand method. The authors could highlight the advantage of the BindFit method compared to the Benesi–Hildebrand method, does the association constant differ if calculated by the Benesi–Hildebrand method?
Answer:
We used BindFit method because this calculator is designed to work with classical supramolecular titration data obtained from NMR, UV, Fluorescence and other methods. The calculator operate by accepting a preformatted excel document with host and guest (ligand) concentrations at each titration points, as opposed to Benesi-Hildebrand method which use for calculating the absorption (A) at certain wave length [D. Brynn Hibbert and Pall Thordarson, Chem. Commun., 2016, 52, 12792-12805. P. Thordarson, Chem. Soc. Rev., 2011, 40, 1305-1323].
- The authors could explain in a little more detail (structurally or stereochemically) why Fluorescein in derivative 2 prefers to bind to the indole system and in derivative 3 it prefers to enter between the pillar[5]arene systems. Is there a 2D NOESY NMR cross-peak that corresponds to the interaction of Fluorescein and benzene hydrogens from the pillar[5]arene skeletons, since in Figure 4 for derivative 2 one molecule of fluorescein is also located between the pillar[5]arene skeleton. From the 2D NOESY NMR spectrum, it seems that only the phenylcarboxylic group of Fluorescein interacts with the indole system, while the tricyclic system interacts with the quaternary amine group of derivative 2.
Answer:
Thank you for your careful consideration of the 2D NOESY NMR spectrum.
We would like to point out that Figure 4 is only schematic representation of the formation of various associates based on synthesized pillar[5]arenes. Due to the low stability of the 2+Fluo associates, fluorescein molecule can easily move around the pillar[5]arene ring interacting non‒covalently with different substituents and sampling a huge number of different poses which additionally was proven by computer simulations. Not excluded that fluorescein can be located between the pillar[5]arene cores, but it is not easy to prove this unambiguously on the basis of the 2D NOESY NMR spectrum, since the proton signal H1 of the pillar[5]arene core overlaps with the diagonal cross-peaks what complicates spectrum interpretation.
The two-dimensional spectrum of the 3+Fluo system is characterized by the presence of a larger number of cross peaks between the aromatic protons of the dye and the aliphatic protons of macrocycle 3 comparing to the system 2+Fluo. In this case, we can assume a significant role of hydrogen forces and electrostatic interactions in the binding of fluorescein with pillar[5]arene 3. The assumption that the dye molecule is located precisely in the pseudocavities formed by macrocyclic substituents, and not in the cavity of the core of pillar[5]arene 3, is associated with the absence of peak shifts in the one-dimensional NMR spectra of the mixture compared to the initial components. Hydrogen bonding between amide protons and oxygen-containing fragments of fluorescein can be indirectly confirmed by a significant decrease in the intensity of amide protons in the spectrum in the low-field region, as well as by computer simulation studies.
In addition, the set of methods (fluorescence and UV-vis spectroscopy, NMR, computer simulation studies, DLS and TEM) used by us to establish the structure of associates is in good agreement with each other, which confirms our assumptions.
- The authors could present how the structure of Fluorescein changes depending on the pH.
Answer:
Chemical structures of fluorescein depending on the pH is in Supporting Information (Figure S46).
- The authors could discuss in 1-2 sentences what would be expected with Fluorescein if the base skeleton of pillar[n]arene n were greater than 5, say 10 or 15.
Answer:
It is rather difficult to predict the expected interactions between pillararenes with n equal to 10 and 15. First of all, this is due to the fact that synthesizing such macrocycles seems to be a nontrivial task. Even macrocycles with n equal to 6 and 7 are obtained in significantly lower yields than pillar[5]arene. Therefore, the chemistry of host-guest complexes for these macrocycles has not been studied, which makes it difficult to make any assumptions. Secondly, with an increase in the number of monomeric units in the macrocycle, the shape of the cavity may be distorted, which will also change the affinity for the substrate.

Round 2
Reviewer 2 Report
The authors improved the manuscript and clarified all doubts.
Author Response
Dear Reviewer,
Thank you very much for careful consideration of the manuscript.